# Qualitative evidence synthesis of values and preferences to inform infant feeding in the context of non-HIV transmission risk

**Christopher Carroll**[1]*, **Andrew Booth**[1], **Fiona Campbell**[1], **Clare Relton**[2]

**1** Health Economics & Decision Science Section, School of Health and Related Research (ScHARR), University of Sheffield, Sheffield, United Kingdom, **2** Institute of Population Health Sciences, Queen Mary University of London, London, United Kingdom

* C.Carroll@sheffield.ac.uk

## Abstract

### Background

Breastfeeding is recommended by many organisations, but feeding choices can take on complexity against a backdrop of a transmissible infection risk. The aim of this synthesis is to explore what is known about the values and preferences of pregnant women, mothers, family members and health practitioners, policy makers and providers (midwives) concerning feeding when there is a risk of Mother-to-Child transmission [MTCT] of an infectious disease (other than HIV/AIDS) to infants (0–2 years of age).

### Methods

A qualitative evidence synthesis and GRADE CERQual assessment of relevant studies of values and preferences regarding infant feeding options in the context of non-HIV MTCT risk.

### Results

The synthesis included eight qualitative studies. Four studies focussed on human T-cell lymphotropic virus type 1 (HTLV-1), three studies on Ebola, and one study on influenza vaccination. Mothers reported feeling sadness and guilt at not breastfeeding, while recognising that it was important for the health of their baby not to breastfeed. Mothers were reportedly appreciative of the provision of appropriate facilities, and the advice of those health professionals who knew about the diseases, but felt other professionals lacked knowledge about the transmission risk of conditions such as HTLV-1. All groups expressed concerns about social perceptions of not breastfeeding, as well as the alternatives. The evidence was coherent and relevant, but there were serious concerns about adequacy and methodological limitations, such as potential social desirability bias in some studies.

### Conclusions

This synthesis describes the reported values and preferences of pregnant women, mothers, and others concerning feeding when there is a risk of Mother-to-Child transmission (MTCT)

**Data Availability Statement:** All relevant data are within the manuscript and its Supporting Information files.

**Funding:** This work was commissioned from the University of Sheffield, UK by the Department of Nutrition and Food Safety, World Health Organization, Switzerland as a technical document to support WHO recommendations on infant feeding.

**Competing interests:** The authors have declared that no competing interests exist.

of an infectious disease (other than HIV/AIDS) to an infant when breastfeeding. However, the evidence in the peer-reviewed literature is limited both in quality and quantity.

## Background

Undernutrition is associated with an estimated 2.7 million child deaths annually or 45% of all child deaths [1]. Infant feeding is key to improved child survival and promotes healthy growth and development. The first two years of a child's life are particularly important, as optimal nutrition during this period lowers morbidity and mortality, reduces the risk of chronic disease, and fosters better development overall. Optimal breastfeeding has been estimated to to save more than 820,000 children under the age of 5 years each year [2]. Early initiation of breastfeeding within 1 hour of birth; exclusive breastfeeding (EBF) for the first 6 months of life; and introduction of nutritionally-adequate and safe complementary (solid) foods at 6 months, together with continued breastfeeding up to 2 years of age or beyond, is recommended (WHO, UNICEF). However, many infants and children do not receive optimal feeding. For example, only an estimated 36% of infants aged 0–6 months worldwide were exclusively breastfed over the period of 2007–2014 [2]. Mothers face many challenges when making the decision whether to initiate or maintain breastfeeding. Pressures exist at an individual, family or community level and also occur during interactions with health service providers.

The decision whether or not to breastfeed may also take place against a backdrop of prevalent risk of disease transmission. In such circumstances, considerations extend beyond the health benefits of breastfeeding, or the risks and advantages of different methods of infant feeding. The decision must now also take into account the level of risk from, and the availability and acceptability of, alternatives to breastfeeding. In general terms, for a breastfeeding mother, infection can occur through many modes of transmission, while for the breastfeeding child infection can also be transmitted, with some diseases, via direct contact vertical transmission (through the breastmilk). Evidence is therefore required to inform an understanding of the values and preferences of pregnant women and mothers, together with others influencing or affected by decisions related to infant feeding [e.g. family members and health practitioners, policy makers and providers, e.g. midwives] in the context of a disease transmission risk from breastmilk. Other than HIV/AIDS, a range of diseases are known to present a risk of mother-to-child transmission (MTCT) (and child-to-mother). These include (but are not limited to): Cytomegalovirus (CMV); Ebola; Herpes Simplex Virus (HSV); Hepatitis; and Rubella [3]. It is therefore important to understand factors that influence decision-making in the context of transmissible illness, and to explore stakeholders' values and preferences, that is, their beliefs, fears, perceptions and experiences around infant feeding. A substantial body of qualitative research, including multiple published systematic reviews and qualitative evidence syntheses (QES), has explored the values and preferences of stakeholders concerning infant feeding within the specific context of the risk of transmission of HIV/AIDS [4–15] and updated WHO guidance has recently been published for this group [16]. However, no systematic review or QES has been performed to date to understand the values and preferences of relevant stakeholders concerning infant feeding when there is the risk of transmission of diseases other than HIV/AIDS.

The WHO therefore commissioned this QES to explore the values and preferences of pregnant women, mothers, family members, health practitioners and providers (midwives), and policy makers, concerning infant feeding, that is, breastfeeding and its alternatives, when there is a risk of Mother-to-Child transmission (MTCT).

## Methods

We conducted a qualitative evidence synthesis in accordance with current best methodological practice, and reported this according to PRISMA-derived ENTREQ guidelines [17, 18].

### Reflexive note

In keeping with quality standards for rigour in qualitative research, the review authors considered how their views and opinions on infant feeding might influence decisions made in the design and conduct of the review. Furthermore, they considered how the emerging results of the study influenced those views and opinions. All authors believed, in line with WHO guidance, that breastfeeding is the preferred method of infant feeding whenever possible, both on health grounds and, in low-and-middle income countries, for resource related reasons. All believed that positive infant feeding experiences are important for the wellbeing of the mother, baby, and the family, in the short and longer term. We therefore used refutational analytic techniques to minimise the risk that these prior beliefs would skew the analysis and the interpretation of the findings [19]. These techniques, initially outlined in the context of meta-ethnography, seek to create opportunities for identifying the "disconfirming case" and include introducing different levels of familiarity of the data and disciplinary perspectives and challenging hierarchical and power relations within the research team [19].

### Protocol and registration

This review was originally conceived as a review of *Acceptable medical reasons for use of breast-milk substitutes* in the context of transmissible disease. The protocol was registered with the PROSPERO CRD database. It is published and available at PROSPERO 2019 CRD42019143387.

### Inclusion criteria

To be included in the review and synthesis, studies were required to satisfy the criteria as outlined in Table 1.

**Table 1. Inclusion criteria, defined using the PerSPEcTiF(S) framework.**

| | |
|---|---|
| Perspective(s) | Women, mothers, partners, carers and significant others, healthcare providers, policy makers |
| Setting | Any setting (primarily community settings) |
| Phenomenon of interest | Infant feeding in the context of the risk of transmission of diseases* other than HIV/AIDS |
| Environment | International, particularly Low- and Middle-Income countries (LMICs) where transmissible diseases are more prevalent |
| Comparison | [Implicitly compared with values and preferences concerning infant feeding where there is no transmission risk or the risk of HIV/AIDS transmission] |
| Timing | When contemplating, carrying-out or supporting breastfeeding, breast milk feeding or alternative infant feeding |
| Findings | Values and preferences: fears, perceptions, experiences and beliefs regarding the phenomenon of interest |
| Study Design | Qualitative studies and Mixed Method studies with a discrete qualitative component. Surveys with qualitative data as free text responses to survey questions were excluded |

*The infectious diseases of interest included, but were not limited to: Chikungunya Virus, Cytomegalovirus, Dengue Fever, Ebola Virus, Hepatitis, Herpes, Influenza, Parvovirus, Rubella, Tuberculosis, Viral Haemorrhagic Fever, West Nile Virus, Yellow Fever, Zika Virus. Human Immunodeficiency Virus (HIV) was included in the search and overall project, but studies exclusively focusing on HIV were excluded from the specific review. It was felt that the literature on HIV and breastfeeding would saturate the review findings at the expense of specific insights relating to other infectious diseases.

## Information sources and search strategies

We searched the following databases for relevant published and unpublished literature from 2000 up to and including March 2019: PubMed; MEDLINE (Ovid); PsycInfo (Ovid); CINAHL (Ovid); EMBASE (Ovid); Web of Science; SCIELO; Scopus; LILACS (for studies conducted in South America); BIREME; African Journals Online (for studies conducted in Africa); and African Index Medicus. Search results were not restricted by publication type. Non-English language studies were included where a translation was available or was possible. Using guidelines developed by the Cochrane QIMG for searching for qualitative evidence [20, 21], search strategies were developed for each database. The search combined thesaurus and free-text terms for transmissible disease and an extensive list of transmissible diseases (see above), with terms for infant feeding, breastfeeding and breast milk, and published filters to identify qualitative research [22, 23]. The search strategy for MEDLINE is available in S1 Table. MEDLINE search. No geographic restrictions were imposed on the search; the date range was limited to 2000–2019 to capture recent and contemporary views. The reference lists of all the included studies and the key publications (i.e. any relevant systematic reviews) were all searched. In addition, citation searches were performed on Google Scholar for each of the included articles. Citation updates were initiated on Google Scholar for all included studies to identify newly published studies published up until the final analyses in November 2019. Further studies were retrieved for inclusion as soon as they were identified.

## Study selection, extraction and appraisal

Using the inclusion criteria, preliminary study screening of all titles and abstracts was conducted by one reviewer (AB) to identify potentially relevant papers, and full text screening of the results was conducted independently by at least two reviewers (FC, AB or CC). A data extraction form was developed and piloted by three reviewers (CC, AB, FC). Two reviewers (CC, FC) then independently performed data extraction and quality assessment (using the CASP Checklist: 10 questions to help you make sense of a Qualitative research) of all included studies using the agreed form; any inconsistencies were resolved by discussion and, if necessary, consultation with a third reviewer (AB).

## Synthesis

A 'best fit' framework synthesis was performed [24–26]. The synthesis used relevant domains from the WHO/UNICEF conceptual model of infant feeding as the theoretical framework to categorise and organise the relevant data [27]. The WHO/UNICEF model depicts conditions that influence feeding decisions and their outcome (e.g., knowledge, perceptions, family influences, resources, environment, etc), as well as issues that need to be explored in order to define appropriate feeding options. The model assumes an ecological perspective, which recognises that feeding behaviours are influenced by interacting, intrapersonal, social and cultural, and physical environment variables. The key concepts in the developed a priori framework were as follows:

- Health system factors;

- Factors relating to the individual;

- Family and community-related factors;

- Socio-economic factors

## Results

Details of the study selection process are presented in Fig 1. The search retrieved 5185 potentially relevant records. 4085 papers were excluded at title and abstract stage, and a further 197 papers were excluded at the full text stage.

The reasons for exclusion of these papers were as follows: 189 were HIV only studies; one was not a qualitative study [28]; two were abstracts only, with incomplete reporting of qualitative findings [29, 30]; and five reported no qualitative evidence on infant feeding and infection transmission risk [31–35] (see S2 Table).

Eight publications satisfied the inclusion criteria (see Table 2). Six were identified from the principal search of bibliographic databases, while two additional papers were identified by citation searching/monitoring [36, 37]. Two studies were conducted in Brazil, with findings

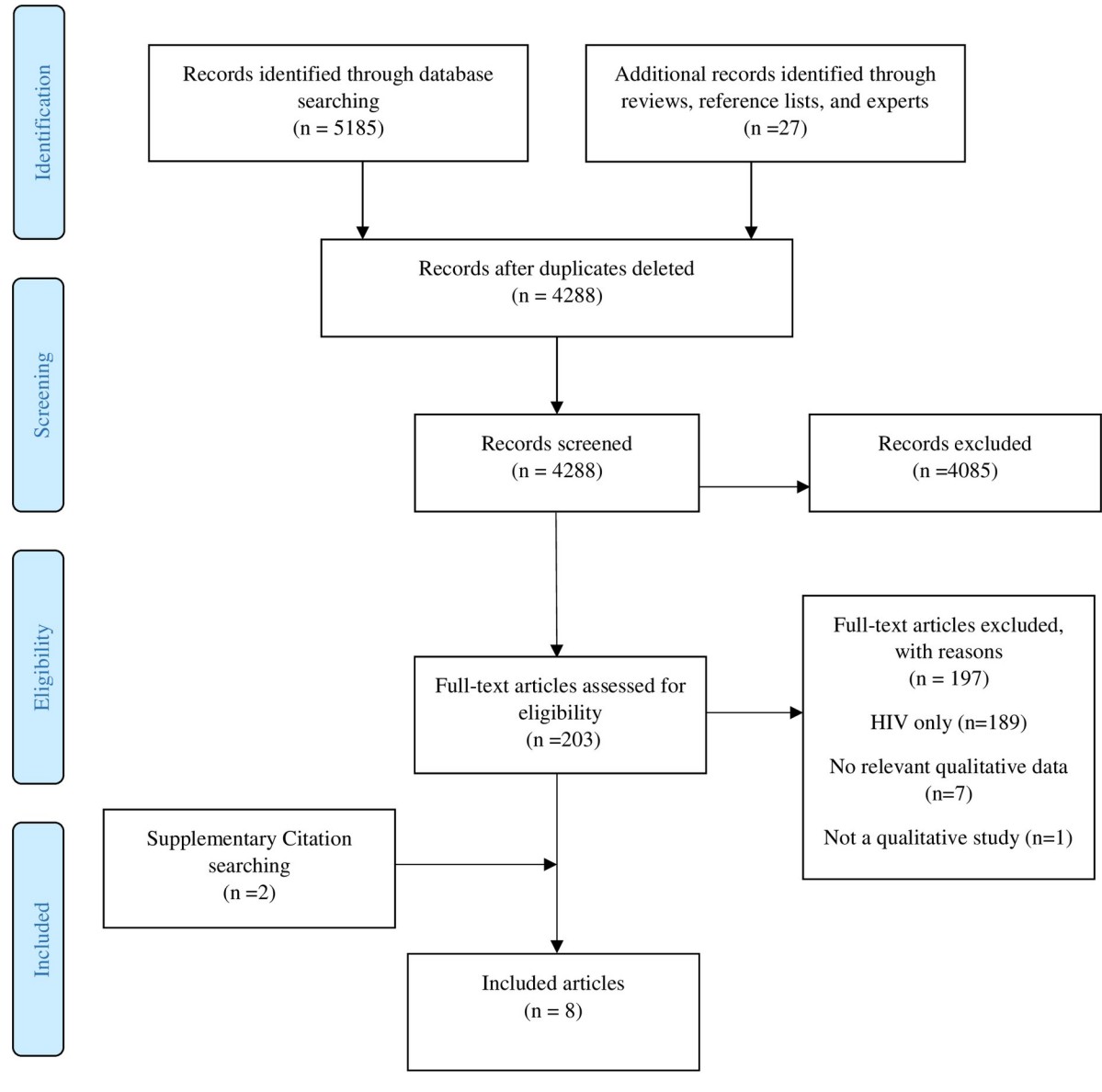

**Fig 1. PRISMA flow diagram.**

**Table 2. Characteristics of included studies.**

| Author (Date) | Setting (i.e. Country) | Vicinity (i.e. Region, State, Province, City) | Study aims and purpose | Infections Included | Perspectives | Sample | How was the sample selected? | Data collection methods used? |
|---|---|---|---|---|---|---|---|---|
| Kodish (2018) [40] | Guinea | Not reported: From 5 of Guinea's 8 administrative regions that were most impacted by Ebola. | First, to understand how the Ebola outbreak may have impacted infant and young child nutrition in Guinea. Second, to understand how stakeholders at multiple different levels perceived the acceptability and effectiveness of the nutrition-specific response during the Ebola outbreak to draw lessons learned and make recommendations for consideration in future similar scenarios. | Ebola Virus | Community, Midwives, Health Providers, Health Managers / Decision-makers, Government Officials / Civil Servants, International Organisations or Agencies. | n = 27 (11 key informants from diverse bodies, including those of the Government /Policy, United Nations, Hospital Management, and Non-Governmental Organizations (NGOs); 6 front-line health workers, 6 household and 4 community members). | Purposive | Interviews |
| Kodish (2019a) [41] | Guinea and Sierra Leone | Not reported | To generate multiple stakeholder perspectives for understanding the nutrition challenges faced during the Ebola virus disease outbreak, as well as for consensus building around improved response strategies. | Ebola Virus | Government Officials/Civil Servants, International Organisations or Agencies. | n = 36 (17 from Guinea, 19 from Sierra Leone [including 4 Ebola survivors]). | Purposive | Interviews, Participatory workshops |
| Kodish (2019b) [37] | Sierra Leone | Across all four provinces | To explore how and through what pathways the Ebola Virus Disease (EVD) outbreak impacted nutrition in Sierra Leone. To investigate the factors to effective implementation of nutrition response strategies during the EVD outbreak. | Ebola Virus | Phase 1: Government hospital managers, government policy makers. Managers working with NGOs or United Nations organisations involved in the outbreak response at the national level. | n = 42 (n = 21 in Phase 1, n = 21 in Phase 2). | Purposive sampling based on role and geographic representation. | Semi-structured interviews |
|  |  |  | To use findings to consider a nutrition preparedness and response framework in planning for future outbreaks. |  | Phase 2: EVD survivors, community leaders, health workers. |  |  |  |
| Teixeira (2017) [36] | Brazil | Salvador, Bahia | To know the feelings of HIV- and HTLV-positive women towards non-breastfeeding. | Human T-cell lymphotropic virus type 1 (HTLV-1) and HIV. | Women. Age ranged from 22 to 86 years, registered at the Reference Center for HTLV; >18 years of age; and having been pregnant at some point. | n = 64 (HTLV-seropositive [HTLV SP] adult women). | Convenience | Descriptive survey with open-ended questions delivered by interview |

*(Continued)*

**Table 2.** (Continued)

| Author (Date) | Setting (i.e. Country) | Vicinity (i.e. Region, State, Province, City) | Study aims and purpose | Infections Included | Perspectives | Sample | How was the sample selected? | Data collection methods used? |
|---|---|---|---|---|---|---|---|---|
| Zihlmann (2017) [39] | Brazil | Sao Paulo | To understand the meanings of inhibiting breastfeeding as a way to prevent the vertical transmission among women living with HTLV-1 and, in addition, to present related situations on experiences of actually interrupting breastfeeding. | HTLV-1. | Mothers, Partners of Women/ Mothers, Infected women without children. | n = 13 (11 women and mothers, 2 men [fathers and/ or partners]). | Convenience | Interviews, Observation |
| Zihlmann (2013) [38] | Brazil | Sao Paulo | To discuss the reproductive decisions of women and men living with HTLV-1 infection, to determine their perception of infection and associated disease and expectations regarding Mother-to-child-transmission (MTCT), and, finally, to assess if health care conditions affect their reproductive decisions. | HTLV-1. | Pregnant Women, Mothers, Partners of Women/ Mothers. | n = 13 (13 HTLV-1-seropositive adults: 11 women and 2 men without co-infections). | Convenience | Interviews |
| Oni (2006) [42] | French Guiana | Maripasoula and Papaıchton | To assess the awareness of human T-cell lymphotropic virus (HTLV) transmission, especially through breastfeeding. | HTLV. | Women with HTLV and some health workers. | n = 40 (40 mothers and women with HTLV type 1; 36 had had children; 29 (average age 39.2 years) had either heard of HTLV or had been told by a doctor that they had a 'blood virus'). | Convenience | Interviews, Questionnaire Survey |
| Nawa (2016) [43] | Japan | NA | To categorize questions by conducting detailed qualitative analyses from the clinicians' viewpoint and to investigate how public concerns regarding influenza vaccinations change over time, particularly in relation to seasonal influenza epidemics. | Influenza, Vaccinations. | The dataset was not limited to any population group. | The 1950 questions used in the detailed analysis were posted by 1684 contributors. | Questions extracted on influenza after excluding those related to avian influenza. | Analysis of data collected from an internet bulletin board of questions and answers. |

EVD: Ebola Virus Disease; HIV: Human Immunodeficiency Virus; HTLV-1: Human T-cell Lymphotropic Virus Type 1; NA: Not applicable; NGO: Non-Governmental Organization.

published in three papers [36, 38, 39]; four studies were conducted in African countries: Guinea [40], Sierra Leone [37] or Guinea and Sierra Leone [41] (the degree of overlap between these two 2019 studies is unclear) and French Guiana [42]; and one in Japan [43]. Three studies (four papers) investigated participants with human T-cell lymphotropic virus type 1 (HTLV-1) [36, 38, 39, 42]; three studies examined Ebola [37, 40, 41], and one explored influenza vaccination [43]. Four papers focused principally on women, mothers and pregnant women with HTLV-1 [36, 38, 39, 42]; one study included a small number of men in its sample [38, 39]. The three Ebola studies exclusively explored the views of health providers, managers, government officials, civil servants and staff from international agencies [37, 40, 41]. The study conducted in Japan did not specify the demographics of the sample [43].

The evidence base was at moderate risk of bias (see S1 Fig): the majority of studies presented a clear question, qualitative design, used appropriate methodology, and reported clear findings; however, only the three studies by Kodish and colleagues [37, 40, 41] clearly reported rigorous recruitment and data collection and analysis strategies; and only one study adequately addressed the relationships between researchers and participants (reflexivity) [37]. These serious methodological limitations were considered within the GRADE-CERQual assessment. Full tables of the GRADE-CERQual Evidence profile for each domain are available in the S3–S6 Tables.

The basic characteristics of the eight included studies are reported in Table 2. The findings of the synthesis, with evidence-based themes presented under the key concepts derived from the *a priori* framework for organising the evidence [27], are presented below. Study findings are reported in the following order, based on weight of evidence, where relevant: HTLV-1, Ebola and the influenza vaccine.

## Factors relating to the individual

**Lack of knowledge among lactating women about risk of Mother-to-Child Transmission (MTCT) by breastfeeding.** Some studies found lack of knowledge, among pregnant women, new parents, front-line workers and caregivers, to represent an important issue. A study on HTLV-1 found that most young women and mothers had not heard of HTLV-1 and were unaware of the risk of transmission from breastfeeding [42].

Within the context of an Ebola outbreak, front-line workers lacked guidance on how to advise lactating mothers [37], they also had to balance conflicting health messages: to promote exclusive breastfeeding and yet to instruct infected mothers to no longer breastfeed. Early in the outbreak, exclusive breastfeeding practices largely continued, but were later replaced by separating sick mothers from their infants.

Nawa *et al.* reported a sharp increase in pregnant and lactating women posting questions about MTCT on a bulletin board when the influenza vaccination became available, and that responses to these posts could be inaccurate [43].

**Power of experts and specialist information and advice.** The influence of specialist health staff could determine both a mother's knowledge and her beliefs. This issue was reported in HTLV-1 studies. One study reported that women and new mothers 'bonded' with specialised infectious diseases centres that gave them advice on infant feeding and how to reduce the risk of HTLV-1 transmission, finding that almost all new mothers trusted and followed that advice, even when faced with contradictory views from other health staff, whom they challenged [38, 39, 42].

**A sense of control.** Once such a bond of trust has been forged, new mothers reported in HTLV-1 studies that a greater sense of autonomy when undertaking difficult decisions impacting on the health of their child [38, 39]. Some participants also reported that the specialist

advice had given them some control over the situation, and that they knew the choice they were making not to breastfeed was of benefit to the health of their child; it was seen as a positive action [39].

**Maternal expectations and bonding.**   In the HTLV-1 studies, mothers consistently reported that they experienced anxiety, sadness, fear and guilt around not being able to breastfeed: mothers' expectations were that breastfeeding was necessary if they were to bond with their baby [36, 38, 39]. Sadness was the overwhelming emotion reported by multiple mothers in one study because they were advised not to breastfeed on account of the risk of transmission [36]; one participant in a study also reported that the situation was difficult for the father [39].

In an Ebola study, health decision-makers and providers reported that it was difficult to challenge the prevailing expectation to breastfeed and, therefore, to implement different or contradictory guidelines on infant feeding [40].

**Stigma.**   Mothers consistently reported in the HTLV-1 studies that they knew little about HTLV-1 and felt a perceived stigma from the requirement that they do not breastfeed [38, 39]. They associated it with HIV because at the time of the studies both conditions required a mother not to breastfeed her baby [38, 39, 42]. In one study a participant reported that she would rather have HIV because at least people knew what it was and were aware of transmission risk [42].

In one Ebola study, it was found that stopping exclusive breastfeeding was not seen as socially acceptable, and that this needed to be addressed during the outbreak in Sierra Leone [37].

## Maternal health

The health of mothers themselves was also an issue identified in three studies. In one HLTV-1 study, mothers reported physical discomfort in their breasts because of milk production and that they could not breastfeed [36].

Two Ebola studies reported findings relevant to maternal health. In one of the studies, participants reported that, if they felt sick, then they were told not to have any contact with their infants, and not to breastfeed infants if they had survived Ebola [41]. Mothers were also reported to be experiencing poor nutrition themselves, as the Ebola outbreak had disrupted the production, availability and access to foods [37].

## Family and community-related factors

**Community views.**   Two studies in HTLV-1 mothers reported that prevalent community views about breastfeeding perceived that failure to breastfeed indicated contagion or HIV infection [38, 39].

This idea that alternatives to breastfeeding were not trustworthy was also reported in an Ebola study [40]. Community 'fear and distrust' was reported by health decision-makers and providers to be an initial barrier to the implementation and uptake of infant feeding alternatives. They further reported that 'intensive and appropriate sensitization efforts increased the acceptability of the food assistance over time' [40].

## Health system factors

**Lack of knowledge among non-infectious diseases staff about risk of Mother-to-Child Transmission (MTCT) by breastfeeding.**   The women and mothers in the HTLV-1 studies consistently reported that the knowledge of health staff without infectious diseases training, including midwives and gynaecologists, was limited; they were uninformed about HTLV-1 and its risks [38, 39, 42]. As a result, health staff pressured new mothers to breastfeed. Staff lack of

knowledge shaped their attitudes and reactions: staff responded with 'disbelief' to statements from new mothers who had been informed about the risks associated with breastfeeding with HTLV-1 [38, 39, 42]. As noted elsewhere, two studies reported that, where pregnant women were diagnosed with HTLV-1, with a risk of transmission by breastfeeding, the women largely accepted and applied specialist information and advice, even to the point of challenging non-specialist infectious diseases staff when they questioned this information [39, 42].

This finding was also reported in two of the studies concerning Ebola. One Ebola study reported that participants felt that community-level training was important for establishing trust, so that the communication of information regarding infant feeding could challenge established practices [40]. The evidence suggests that within the context of an Ebola outbreak, behaviour change in response to information giving can occur and lead to sustained change where the communications are multi-channelled, include numerous forms of print, interpersonal and mass media, and are culturally appropriate and understandable [37]. Community involvement was important for designing an effective, respectful and well-planned response.

**Appreciation of facilities to support private infant feeding.** The provision of appropriate facilities for infant feeding within hospitals was welcomed (e.g. new mothers with HTLV-1 reported relief at having a separate room so that they did not have to watch other new mothers breastfeed, and were themselves not observed not to be breastfeeding) [39].

**Provision of trustworthy alternative feeding options.** None of the studies on HTLV-1 reported findings on this theme, but all three Ebola studies did so. Access to and the reliability of replacement feeding options was also seen as important: health decision-makers and providers reporting that availability improved when more resources were provided, and that replacement feeding was seen as more reliable as trust in authorities increased [40]. Health decision-makers and providers also reported that outbreaks of infectious disease (Ebola) in a community only caused feeding practices to change gradually as information was communicated about transmission, but it also required the community to gain trust in the infant feeding alternatives, and for sufficient alternatives to be available [41]. Access to replacement feeding, and complementary feeding, were also disrupted by the Ebola outbreak meaning that infants and young children were disproportionately affected [37]. In the context of the Ebola outbreak, many children were orphaned and members of the wider, extended family had to become carers and then take on the responsibility for providing replacement feeds for breastfed infants [37]. Some children who were used to breastfeeding took time to adjust to using the ready-to-use infant formulas. Food distribution that is co-ordinated and well planned, and can accommodate a surge in new international organisations arriving to assist, was reported to be crucial to ensuring access to replacement and complementary foods for infants [37].

### Socio-economic factors

**Cost of alternatives to breastmilk can be prohibitive.** One study of women with HTLV-1 reported that women responded to the cost of artificial milk by diluting these alternatives [42].

The economic impact of the Ebola outbreak in Sierra Leone was found to exacerbate existing food insecurity [37]. Food availability and access presented a serious challenge during the outbreak, many families were unable to farm, and/or work. Coupled with higher demand and higher prices, access to complementary feeds was affected, and so too were infant feeding practices.

### Discussion

The key findings of the synthesis, and the confidence in these findings (based on the results of the GRADE CERQual assessment of the evidence, see Table 3) might be summarised as follows:

**Table 3. GRADE-CERQual summary of qualitative findings.**

| Summary of review finding | Studies contributing to review finding | GRADE-CERQual assessment of confidence in the evidence | Explanation of GRADE-CERQual assessment |
|---|---|---|---|
| Factors relating to the individual | | | |
| Lactating women lack knowledge about risk of transmission of HTLV-1 and influenza vaccination from mother-to-child by breastfeeding | [37, 42, 43] | Low | Three studies (French Guiana, Japan, Sierra Leone). There are moderate concerns about coherence, and serious concerns about methodological limitations, adequacy and relevance (one study was HTLV-1 and one on influenza vaccination). |
| New mothers were strongly influenced by the information and advice on mother-to-child transmission (MTCT) provided by specialist infectious diseases health staff. New mothers feel empowered by this information and advice | [38, 39, 42] | Moderate | Three studies (two Brazil, one French Guiana). Moderate concerns about methodological limitations, coherence, adequacy and relevance (all studies only consider HTLV-1). |
| New mothers report that when information and advice is given by health staff with specialist expertise, this gives them confidence in their choices | [38, 39] | Moderate | Two studies (both Brazil). Moderate concerns about methodological limitations, coherence, adequacy and relevance (all studies only consider HTLV-1) |
| New mothers maintain strong expectations about the need to breastfeed if they are to form bonds with their baby | [36, 38–40] | Moderate | Four studies (three Brazil, one Guinea). Minor concerns over coherence, and moderate concerns about methodological limitations, relevance and adequacy. |
| Mothers experience stigma as a consequence of not being able to breastfeed | [37–39, 42] | Moderate | Four studies (two Brazil, one French Guiana, one Sierra Leone). There are moderate concerns about methodological limitations, coherence, adequacy and relevance (all studies only consider HTLV-1). |
| Mothers' health can affect their ability to breastfeed | [36, 37, 41] | Low | Three studies (Guinea, Sierra Leone, Brazil). There are moderate concerns about methodological limitations and coherence, and serious concerns over adequacy and relevance. |
| Community-related factors | | | |
| Health decision-makers and managers reported a prevalent view in the community that failure to breastfeed indicated contagion or infection | [38, 39] | Low | Two studies (both Brazil). Minor concerns over coherence, moderate concerns about methodological limitations, but serious concerns about adequacy and relevance (e.g. HTLV-1 and Brazil only). |
| According to health decision-makers and managers, those in the community believed that alternatives to breastfeeding were not trustworthy. | [40] | Low | One study (Guinea and Sierra Leone) of a single condition (Ebola). Minor concerns about methodological limitations and coherence, but serious concerns about adequacy and relevance. |
| Health system factors | | | |
| Women and new mothers report a lack of knowledge among non-infectious diseases health staff about certain conditions with a risk of MTCT by breastfeeding [e.g. HTLV-1] | [37–40, 42] | Moderate | Five studies (two Brazil, one each French Guiana, Sierra Leone, Guinea). Minor concerns about coherence, and moderate concerns about methodological limitations, adequacy and relevance (all studies only consider HTLV-1 or Ebola, and only from a single perspective) |
| New mothers appreciate facilities that provide privacy for infant feeding because they are not exposed to observation by others and therefore are less likely to experience stigma from being identified as having a transmissible disease | [39] | Low | One study (Brazil) with minor concerns about coherence, moderate concerns about methodological limitations, but serious concerns about adequacy and relevance (e.g. only HTLV-1) |
| Health decision-makers and managers report that establishing trust between providers and mothers is important if established practices on infant feeding are to be successfully challenged when there is a disease outbreak [e.g. Ebola]. | [37, 40, 41] | Low | Three studies (Guinea, Sierra Leone). There are minor concerns about methodological limitations, moderate concerns about coherence, and serious concerns over adequacy and relevance. |

(*Continued*)

**Table 3.** (Continued)

| Summary of review finding | Studies contributing to review finding | GRADE-CERQual assessment of confidence in the evidence | Explanation of GRADE-CERQual assessment |
|---|---|---|---|
| Health decision-makers and managers report that it is important for alternatives to breastfeeding to be available and trustworthy if established practices of exclusive breastfeeding [EBF] are to be challenged. | [37, 40] | Low | Two studies (Guinea and Sierra Leone) of a single condition (Ebola). There are minor concerns about methodological limitations, moderate concerns about coherence, and serious concerns over adequacy and relevance. |
| Socio-economic factors | | | |
| Mothers report that the cost of alternatives to breast-milk can be prohibitive. | [37, 42] | Low | Two studies (French Guiana, Sierra Leone) of HTLV-1 and Ebola. There are minor concerns about coherence, moderate concerns about methodological limitations in one study, and serious concerns about adequacy and relevance. |

- The community, specifically women, new mothers and frontline health staff, report that they lack knowledge regarding MTCT of Ebola and HTLV-1 via breastfeeding. Confidence in this finding that they lack knowledge is very low because of moderate concerns about coherence of the evidence, and serious concerns about methodological limitations, adequacy and relevance (one study was HTLV-1 and one on influenza vaccination).

- New mothers report feeling sad and anxious that their circumstances mean they cannot breastfeed. They also report being deeply concerned that their bond with their child might be affected as a result. However, mothers appear to be willing to learn about the risks associated with these conditions, and to adapt their feeding practices accordingly, provided that specialist infectious diseases advice is communicated in an appropriate and sensitive manner, and trustworthy and affordable alternative feeding options are available. In such circumstances, mothers might implement changes in infant feeding behaviour, even if such changes could mean experiencing stigma and the need to challenge other health staff. The confidence in this finding that women are prepared to adapt is moderate given concerns over adequacy, but the evidence is coherent and relevant.

- New mothers were reported to value accurate infectious diseases advice which would protect their baby, as well as the provision of trustworthy alternatives and facilities in hospitals to enable non-public infant feeding, thereby reducing exposure to any perceived stigma that they might experience from a failure to be seen breastfeeding. The confidence in this finding, that they value such advice, is moderate given concerns over adequacy, but the evidence is coherent and relevant.

- Health providers and decision-makers reported community views that failure to breastfeed indicated contagion; while mothers reported experiencing guilt and stigma for the same reason, and a need not to be seen to go against the community by not breastfeeding. The confidence in this finding of stigma is low given moderate concerns about coherence of the evidence, and serious concerns about methodological limitations, adequacy and relevance linked to lack of evidence–no more than one or two studies each from health providers and from mothers.

- Health providers and decision-makers also reported a lack of confidence among mothers in the alternatives to breastfeeding, and mothers and community members report that access to, and the cost of such alternatives, is prohibitive. Confidence in this finding, that mothers didn't have confidence in alternatives, is low given moderate concerns about coherence of

the evidence, and serious concerns about methodological limitations, adequacy and relevance.

Only two of the transmissible infections of interest (HTLV-1 and Ebola) are covered in the included studies. One group of papers explored the views, predominantly of women, and principally from Brazil, regarding infant feeding and risk of transmission of HTLV-1 [38, 39, 42]. Brazil is the principal location for HLTV-1 [44], so these data are highly relevant. HTLV-1 has been reported to affect 5–10 million people worldwide, with up to 2 million estimated to be in Brazil [44]. The risk of Adult T-cell leukaemia for children who experience MTCT [45] highlights the need to prevent MTCT of HTLV-1. The lack of knowledge about the condition, its implications, and related stigma, exist generally and not only in relation to infant feeding [44]. Findings for this transmissible disease from mothers and families were coherent and relevant, although there were only four studies with a principal focus on mothers. Views of health providers and policy-makers were not reported.

The second group of papers–three related publications lead authored by Kodish—explored the views of health care decision-makers, managers and providers in Guinea and Sierra Leone regarding infant and child nutrition generally following an Ebola outbreak; infant feeding and transmission risk was not the focus of these studies [37, 40, 41]. As a result, while these findings on diverse key individuals' views on infant feeding are important, it must be cautioned that the sample of studies is not high quality and is also highly-localised, so its external validity is extremely limited.

QES is an interpretive approach to synthesis so the findings do not share the more objective levels of interpretation that might apply to some quantitative approaches. However, the path from the published evidence to the summary of findings, and the related decision frameworks, is transparent and auditable. In performing this synthesis, established approaches were used to analyse the evidence and gauge confidence in the findings. Sensitive searching was conducted in seeking to identify all relevant research. Despite extensive searching, the evidence base identified was very limited. Almost no evidence was found that related to socio-economic, cultural or religious factors, or the influence of partners or other family on decision-making. There was also no exploration of antenatal support and limited acknowledgement of issues such as the physical consequences for a lactating mother is being unable to breastfeed. One study is of questionable value: an analysis of qualitative data from a very large number of anonymous participants posting on an internet bulletin board regarding influenza vaccination [43].

The implications of the findings are highly uncertain for other transmissible infections given that there was only limited data from mothers and family for one disease (HLTV-1) and healthcare providers and policy-makers, and the community more broadly, for another (Ebola). Qualitative studies are therefore required on infant feeding and transmission risk in the context of conditions other than HLTV-1 (and HIV/AIDS). Such studies would provide direct evidence of relevance to the production of guidelines, as well as providing indirect evidence to enable an assessment of the transferability of findings from better-researched conditions and contexts, such as HIV/AIDS and HTLV-1. The values and preferences of a broader spectrum of stakeholders should also be explored in qualitative studies.

## Conclusion

The evidence base is limited in quality and quantity. Despite extensive searching for qualitative evidence of values and preferences concerning infant feeding in the context of multiple transmissible diseases other than HIV/AIDS, only two such diseases were identified in any relevant paper. Ebola was covered by three (related) studies, while four of the eight included papers focused on HTLV-1, three of these having been conducted in Brazil. The extent to which the

findings from these studies are transferable to other infectious diseases, and other countries, is uncertain. However, some findings from this synthesis support moderate confidence, in particular mothers' expectations, the consequent stigma and sadness associated with not being able to breastfeed, and the perceived preference for advice and support from infectious diseases health staff. However, the small number of studies, the majority with methodological limitations, has resulted in only low or very low confidence in most findings.

## Supporting information

**S1 Checklist.**
(DOC)

**S1 Fig. CASP quality assessments for studies on values and preferences concerning infant feeding in the context of non-HIV transmission risk.**
(TIF)

**S1 Table. MEDLINE search.**
(DOCX)

**S2 Table. Excluded studies (full-text checked) [principal reason for exclusion, other than HIV/AIDS]).**
(DOCX)

**S3 Table. GRADE-CERQual evidence profile: Factors relating to individuals.**
(DOCX)

**S4 Table. GRADE-CERQual evidence profile: Community related factors.**
(DOCX)

**S5 Table. GRADE-CERQual evidence profile: Health system factors.**
(DOCX)

**S6 Table. GRADE-CERQual evidence profile: Socio-economic factors.**
(DOCX)

## Author Contributions

**Conceptualization:** Christopher Carroll, Andrew Booth, Clare Relton.

**Data curation:** Andrew Booth.

**Formal analysis:** Christopher Carroll, Andrew Booth, Fiona Campbell.

**Funding acquisition:** Christopher Carroll, Andrew Booth, Clare Relton.

**Methodology:** Christopher Carroll, Andrew Booth.

**Validation:** Fiona Campbell.

**Writing – original draft:** Christopher Carroll, Andrew Booth.

**Writing – review & editing:** Christopher Carroll, Andrew Booth, Fiona Campbell, Clare Relton.

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
