## [Decision Letter · Decision Letter 0]

8 Oct 2020

PONE-D-20-28562

Qualitative Evidence Synthesis of Values and Preferences to Inform Infant feeding in the Context of non-HIV Transmission Risk

PLOS ONE

Dear Dr. Carroll,

Thank you for submitting your manuscript to PLOS ONE. After careful consideration, we feel that it has merit but does not fully meet PLOS ONE’s publication criteria as it currently stands. Therefore, we invite you to submit a revised version of the manuscript that addresses the points raised during the review process.

We look forward to receiving your revised manuscript.

Kind regards,

Yukiko Washio, Ph.D.

Academic Editor

PLOS ONE

Journal Requirements:

2. Please update your search to include all studies published in the last 12 month.

Reviewers' comments:

Reviewer's Responses to Questions

**Comments to the Author**

1. Is the manuscript technically sound, and do the data support the conclusions?

Reviewer #1: Yes

Reviewer #2: Yes

2. Has the statistical analysis been performed appropriately and rigorously? 

Reviewer #1: Yes

Reviewer #2: N/A

3. Have the authors made all data underlying the findings in their manuscript fully available?

Reviewer #1: Yes

Reviewer #2: Yes

4. Is the manuscript presented in an intelligible fashion and written in standard English?

Reviewer #1: Yes

Reviewer #2: Yes

5. Review Comments to the Author

Reviewer #1: The authors present a qualitative evidence synthesis to review and summarise current evidence on the values and preferences of pregnant women, mothers, family members, health practitioners, policy makers and providers regarding breast-feeding when there is a risk of transmitting an infectious disease from mother to infant. While there currently are few manuscripts available on this topic, some of which are of limited quality, the authors review, assess and summarise the findings adequately, resulting in adequate conclusions and recommendations.

Some minor issues that should be addressed include:

- It is unclear why the job title is included in the affiliation of the last author

- The abbreviation MTC is introduced several times in the manuscript, yet not used consistently throughout the manuscript, e.g. abstract line 4, page 4/line 17, page 17 etc.

- Fig S3 is of poor quality and most of the text is not readable. Once updated, the manuscript might benefit from including the figure into the main text.

- Table 2 could be structured better to allow the reader to get a clearer picture of the information provided. Further there are some grammatical mistakes, e.g. the fullstops are missing in the perspectives column for study 43.

- The sentence structure is unclear at some parts of the manuscript, e.g. p18/line 20-22 or p22/line 10-11

- It might be helpful to structure the result sections according to the infectious disease that was evaluated in the studies as values and preferences might differ depending on which infectious agent might be transmitted via breastfeeding

Reviewer #2: The QES reports on an important area in the context of infant health. It also specifically addresses an area of breastfeeding and risks of MTC seldom studied compared to MTC transmission as it relates to HIV.

The paper is well written and the methods adequately explained.

The results are well synthesized and findings from the 8 studies well summarized.

Some minor specific comments:

page 2 line 14: replace breast feed with breastfeed

page 2 line20: information such as?

page 4 line 6: the sentence awkward - could be replaced with : Optimal breastfeeding is estimated to save more than 820 000 children.....?

page 4 line 15: replace in with during.

page 6 line 17: i suggest changing sentence to: could you change this to:

To be included in the review and synthesis, studies were required to satisfy the criteria as outlined in Table 1.

page 17 line 24: health staff such as who? in line 20 the use of health staff seem to refer to those trusted and not trusted and here only those not trusted.

page 18 line 20: could this sentence be rephrased?

page 19 line 6-7: could not lactate or could not breastfeed?

page 19 line 14: could this sentence be re-phrased? is this an attitude towards breastfeeding. This sentence is awkward.

6. PLOS authors have the option to publish the peer review history of their article (what does this mean?). If published, this will include your full peer review and any attached files.

Reviewer #1: No

Reviewer #2: No

---

## [Decision Letter · Decision Letter 1]

9 Nov 2020

Qualitative Evidence Synthesis of Values and Preferences to Inform Infant feeding in the Context of non-HIV Transmission Risk

PONE-D-20-28562R1

Dear Dr. Carroll,

We’re pleased to inform you that your manuscript has been judged scientifically suitable for publication and will be formally accepted for publication once it meets all outstanding technical requirements.

Kind regards,

Yukiko Washio, Ph.D.

Academic Editor

PLOS ONE

Additional Editor Comments (optional):

Reviewers' comments:

Reviewer's Responses to Questions

**Comments to the Author**

1. If the authors have adequately addressed your comments raised in a previous round of review and you feel that this manuscript is now acceptable for publication, you may indicate that here to bypass the “Comments to the Author” section, enter your conflict of interest statement in the “Confidential to Editor” section, and submit your "Accept" recommendation.

Reviewer #1: All comments have been addressed

Reviewer #2: All comments have been addressed

2. Is the manuscript technically sound, and do the data support the conclusions?

Reviewer #1: Yes

Reviewer #2: Yes

3. Has the statistical analysis been performed appropriately and rigorously? 

Reviewer #1: N/A

Reviewer #2: N/A

4. Have the authors made all data underlying the findings in their manuscript fully available?

Reviewer #1: Yes

Reviewer #2: Yes

5. Is the manuscript presented in an intelligible fashion and written in standard English?

Reviewer #1: Yes

Reviewer #2: Yes

6. Review Comments to the Author

Reviewer #1: The authors have addressed all issues raised.

On page 5, line 17: MTCT already introduced in previous paragraph

Reviewer #2: (No Response)

7. PLOS authors have the option to publish the peer review history of their article (what does this mean?). If published, this will include your full peer review and any attached files.

Reviewer #1: No

Reviewer #2: No

---

## [Editor Report · Acceptance letter]

19 Nov 2020

PONE-D-20-28562R1 

Qualitative Evidence Synthesis of Values and Preferences to Inform Infant feeding in the Context of non-HIV Transmission Risk 

Dear Dr. Carroll:

I'm pleased to inform you that your manuscript has been deemed suitable for publication in PLOS ONE. Congratulations! Your manuscript is now with our production department. 

Kind regards, 

on behalf of

Dr. Yukiko Washio 

Academic Editor

PLOS ONE